traditional faith healers; biomedical care; low- and middle-income countries; collaborative care; mental illness

**Corresponding author:**
Swaran P. Singh;
Email: s.p.singh@warwick.ac.uk

*The Transforming access to care for serious mental disorders in slums; https://warwick.ac.uk/fac/sci/med/research/hscience/mhwellbeing/projects/transform.

# A scoping review to evaluate the efficacy of combining traditional healing and modern psychiatry in global mental healthcare

Sagar Jilka[1,2,3] , Catherine Winsper[1], Samantha A. Johnson[4], Onaedo Ilozumba[5] , Ryan G Wagner[6], Sanjana Subhedar[1], Dafne Morroni[1] , Richard Lilford[5], Swaran P. Singh[1,3,6] and On behalf of the TRANSFORM consortium*

[1]Warwick Medical School, University of Warwick, Coventry, UK; [2]Institute of Psychiatry, Psychology and Neuroscience, King's College London, London, UK; [3]Warwick Centre for Global Health, University of Warwick; [4]Library Services, University of Warwick, Coventry, UK; [5]Institute of Applied Health Research, University of Birmingham, Birmingham, UK and [6]MRC/Wits Agincourt Research Unit, School of Public Health, Faculty of Health Sciences, University of Witwatersrand, Johannesburg, South Africa

## Abstract

Traditional faith healers (TFHs) are often consulted for serious mental illness (SMIs) in low- and middle-income countries (LMICs). Involvement of TFHs in mental healthcare could provide an opportunity for early identification and intervention to reduce the mental health treatment gap in LMICs. The aim of this study was to identify models of collaboration between TFHs and biomedical professionals, determine the outcomes of these collaborative models and identify any mechanisms (i.e., explanatory processes) or contextual moderators (i.e., barriers and facilitators) of these outcomes. A systematic scoping review of five electronic databases was performed from inception to March 2023 guided by consultation with local experts in Nigeria and Bangladesh. Data were extracted using a predefined data charting form and synthesised narratively. Six independent studies (eight articles) satisfied the inclusion criteria. Study locations included Ghana (n = 1), Nigeria (n = 1), Nigeria and Ghana (n = 1), India (n = 1), Hong Kong (n = 1) and South Africa (n = 1). We identified two main intervention typologies: (1) Western-based educational interventions for TFHs and (2) shared collaborative models between TFHs and biomedical professionals. Converging evidence from both typologies indicated that education for TFHs can help reduce harmful practices. Shared collaborative models led to significant improvements in psychiatric symptoms (in comparison to care as usual) and increases in referrals to biomedical care from TFHs. Proposed mechanisms underpinning outcomes included trust building and empowering TFHs by increasing awareness and knowledge of mental illness and human rights. Barriers to implementation were observed at the individual (e.g., suspicions of TFHs), relationship (e.g., reluctance of biomedical practitioners to equalise their status with TFHs) and service (e.g., lack of formal referral systems) levels. Research on collaborative models for mental healthcare is in its infancy. Preliminary findings are encouraging. To ensure effective collaboration, future programmes should incorporate active participation from community stakeholders (e.g., patients, caregivers, faith healers) and target barriers to implementation on multiple levels.

## Impact statement

This systematic scoping review of collaborative models between biomedical and traditional practitioners highlights a significant gap in mental healthcare delivery, particularly in low- and middle-income countries. By demonstrating the effectiveness of integrated approaches, this research contributes to a paradigm shift in mental health treatment, emphasising the importance of culturally sensitive practices. The findings underscore that collaboration between traditional healers and biomedical practitioners not only enhances treatment outcomes but also fosters trust and respect within communities. This research advocates for the adoption of such collaborative models on a broader scale, encouraging policymakers and healthcare systems to recognize and integrate traditional healing practices. Ultimately, this work aims to improve mental health access and reduce stigma, contributing to a more inclusive and holistic healthcare framework that could have far-reaching implications for mental health policy and practice globally.

## Introduction

Twelve per cent of the global disease burden is due to mental and behavioural disorders (World Health Organization, 2001), and more than 70% of this is experienced in low- and middle-income countries (LMICs) (Tomlinson, 2013). The mental health treatment gap (i.e., the difference between the number of people who need care and those who receive it) is between 80% and 93% in some LMICs (WHO World Mental Health Survey Consortium, 2004; National Institute of Mental Health, 2019), indicating that less than 1 in 10 are able to access appropriate care. In most LMICs, public mental health systems do not receive adequate investment (Joarder et al., 2019), and of the overall annual health budget, little is designated for mental health (World Health Organization, 2020).

Help-seeking for serious mental illnesses (SMIs) in LMICs is pluralistic, with traditional and faith-based healers (TFHs) often being the initial, and sometimes only, port of call (Lilford et al., 2020; Farooq et al., 2023; Singh et al., 2023). Traditional or faith-based healing can alleviate mild symptoms of mood and anxiety disorders and provide valued social and spiritual support, but very little evidence exists that traditional practices improve care or outcomes for SMIs (Nortje et al., 2016; Van der Watt et al., 2018). Crucially, reliance on traditional or faith-based systems can lead to harmful treatment practices (i.e., physical restraint, beating, confinement; Esan et al., 2019), longer duration of untreated psychosis and poorer outcomes for people with psychotic illnesses (Lilford et al., 2020).

Limited availability of biomedical mental healthcare in LMICs coupled with concerns regarding harmful treatment practices delivered by traditional healers indicates the need for collaborative models between faith healers and the modern healthcare system to improve accessibility and reduce fragmentation through models of integrated care (Green and Colucci 2020; Singh et al., 2023). Indeed, there is evidence that a combined approach can be successful in the realm of physical health problems including tuberculosis and HIV (Veling et al., 2019). Further, joining modern and traditional approaches could help provide holistic care incorporating the patient's cultural framework (Saha et al., 2021) including their spiritual and religious beliefs, which is an important element of mental healthcare globally (Winsper et al., 2024).

We could not identify any extant reviews on components and/or outcomes of collaborative models for mental healthcare; however, a recent systematic review considered traditional healers' and biomedical practitioners' perceptions of collaborative mental healthcare in LMICs (Green and Colucci 2020). The authors identified 14 studies (13 from Africa) and concluded that while TFHs and biomedical practitioners had different conceptualisations of mental illness, they are willing to work together to provide a holistic service. Building on this work, the aim of the current scoping review is to explore the literature to identify intervention studies on collaborative care models between TFHs and biomedical practitioners for mental illness.

Specifically, we aim to identify (1) the types of available evidence, (2) typologies of collaboration developed between TFHs and biomedical doctors, (3) outcomes of these collaborations and (4) potential mechanisms and contextual moderators underpinning reported outcomes.

## Methods

We conducted a systematic scoping review as the literature on outcomes of collaborative interventions has not been previously reviewed, and our initial exploration indicated a heterogeneous body of literature (Peters et al., 2015). The current review is part of our NIHR-funded global mental health project (TRANSFORM) to improve outcomes of people with SMI in Nigeria and Bangladesh (Singh et al., 2022) and will help inform an innovative collaborative care model between TFHs and mental health professionals. As recommended by Peters et al. (2015), we developed an *a priori* scoping review protocol in collaboration with local stakeholders from Nigeria and Bangladesh. The protocol included details on objectives, methods and proposed plans.

### Eligibility criteria

The PICO model of Miller and Forrest (2001) was applied as the search strategy tool for this scoping review.

Population (P): We included studies that focused on participants from formal and informal settings (e.g., formal: psychiatrists, Community Health Worker (CHWs); informal: traditional, faith, religious healers, drug sellers). We defined healers as "healers who explicitly appeal to spiritual, magical or religious explanations for disease and distress" (Nortje et al., 2016).

We excluded the qualitative viewpoint or outcomes from the perspective of the persons with lived experience and their caregivers in order to focus on the outcomes of potential interventions. This included the effects of personal religiosity and spirituality, so-called distant healing where the patient is not directly involved in the intervention, and Western psychotherapies that incorporate religious elements.

Intervention (I): The intervention can include care provided by TFHs (under the definition of TFH as given above). It can include any traditional or faith-based intervention provided by TFHs independently or any evidence-based treatment on which traditional healers were trained by biomedical/mental health professionals or any care provided by both traditional and biomedical professionals in collaboration. However, studies were excluded if traditional healers provided any oral or topical or nasal or inhaling herbal/chemical/substances for the management of common mental illness.

We included interventions where a collaboration between the sectors did not directly investigate patient outcomes, but the collaboration aimed to improve TFHs' knowledge, attitudes and practices towards mental health.

We included studies providing quantitative data on a treatment-seeking population for mental disorder or quantitative data on TFH outcomes based on any collaboration with the biomedical sector.

Comparison (C): We included all studies in which there was a comparator for sample (population), outcomes and/or where the comparison was related to a change over time. We included studies whose research questions fulfil the current reviews; research questions irrespective of if the study was a control or comparator.

Outcomes (O): We want to understand the primary and secondary outcomes of the above interventions, what instruments were used and how these data were collected and for whom. Studies were included if they provided quantitative data pertaining to the outcomes of a collaborative intervention for mental illness. Regarding the third aim, we focused on qualitative studies investigating the subjective opinions of participants (i.e., Informal and formal staff) about collaborating in the care of people with lived experiences.

Pilot studies, pre-post studies and randomised controlled trials were eligible for selection. Studies had to be published peer-reviewed and to be included in the review. Studies were excluded if they reported duplicate data. Unpublished studies including

dissertation and conference abstracts were excluded. Review articles and qualitative studies (with no complementary quantitative data) were also excluded from the review. To be included in the review, papers had to be written in the English language.

## Search strategy

Following advice from the University's information specialist (SAJ), we searched MEDLINE ALL (OVID, 1946–), Embase (OVID, 1947–), PsychInfo (OVID, 1806–), CINAHL (EBSCO, 1981–), Web of Science (Clarivate, 1900–) to 2 March 2023, and subsequently ran an updated search from 2 March 2023 to 4 December 2024, combining the following three search strings: ("traditional healer" OR "spiritual healer" OR "religious healer" OR diviner OR shaman OR "traditional practitioner") AND ("healthcare professional" OR "healthcare worker" OR doctor OR psychiatrist OR nurse OR psychotherapist) AND ("mental health" OR "mental disorder" OR "mental illness" OR "mental health services" OR "mental healthcare" OR "serious mental disorder" OR "serious mental illness" OR "severe mental illness" OR "severe mental disorder"). Reference lists of all selected articles were searched for additional studies (including those providing additional details on interventions included in the review). Our search strategy can be found in Supplementary Figure 1a and 1b.

## Study selection

SJ screened all returned titles and abstracts to select full-text articles based on the inclusion and exclusion criteria. Two researchers (SJ and OR) independently screened the full-text articles for inclusion in the final review. Disagreements were independently discussed with a third researcher (RW).

## Data-charting process

A data-charting form was developed *a priori* to record details of the included studies. It included first author, year of publication, study location, study design, sample, main assessment tools, intervention description, main findings, potential mechanisms underpinning interventions and contextual moderators (i.e., implementation facilitators and barriers).

## Synthesis of results

Studies were organised according to the typology of intervention (i.e., educational approaches versus collaborative models) and results were presented in tables to assess whether there were any common outcomes, mechanisms or moderators across studies to inform future intervention development.

## Results

### Search results

Figure 1 summarises the search process. We identified 3266 papers from five databases. After removing duplicates (n = 433), 2833 papers were retrieved and screened on their title and abstract. Twenty-five papers were found to meet the criteria at the full-text screening stage. Following screening, seven articles were selected for inclusion in the final review. Agreement between reviewers for final full-text inclusion was 96%. Authors discussed the reasons behind the discrepancy in selected articles, which was related to the

qualitative study design of one of the articles (Yaro et al., 2020). As this study pertained to one of the RCTs (Ofori-Atta et al., 2018) included in the review, the authors agreed to include the article to elicit additional information on intervention components, mechanisms and contextual moderators. An additional article (Shields et al., 2016) was identified through citation scanning of eligible studies. This article provided additional qualitative data on a study identified through the database search (Saha et al., 2021). Thus, there were eight articles (six independent studies) included in the final review.

## Study characteristics

Table 1 provides an overview of study characteristics, intervention components and main results. Studies comprised a range of research designs including pre-post studies (n = 2), cluster randomised controlled trial (n = 1), randomised controlled trial (n = 1), secondary analysis (n = 1), mixed methods study (n = 1), pilot study (n = 1) and qualitative evaluation of an RCT (n = 1). Some interventions focused on improving TFHs' mental health knowledge, practice, identification (Adelekan et al., 2001; Lam et al., 2016) and referral skills (Veling et al., 2019), others focused on the management of psychotic (Gureje et al., 2020) or schizophrenic/mood disordered (Ofori-Atta et al., 2018; Yaro et al., 2020; Saha et al., 2021) patients through collaborative models between traditional and biomedical practitioners. We categorised interventions into two broad typologies: (1) Western-based information, education, and communication (IEC) interventions for TFHs; and (2) shared collaborative models between TFHs and biomedical professionals. It should be noted there was a degree of overlap between typologies (e.g., some collaborative models included training for traditional healers).

## Terminology

Studies used different terms to describe traditional healers, including traditional mental health practitioners (TMHPs; Adelekan et al., 2001), traditional faith healers (TFHs; Ofori-Atta et al., 2018; Gureje et al., 2020), traditional Chinese medicine (TCM) practitioners (Lam et al., 2016), FBHs (Saha et al., 2021), TMHPs (Veling et al., 2019) and spiritual healers (Yaro et al., 2020). We retain these terms in our description of the studies below.

## Synthesis of results

First, we present the main results of each study including components of interventions and outcomes (Table 1). Next, we present any available data on potential mechanisms and contextual moderators of the interventions (Table 2).

### (1) Western-based IEC interventions for TFHs

Three studies evaluated interventions designed to train and/or educate TMHPs in Western mental health principles and practices to increase awareness, knowledge, identification and referral skills. Two were pre-post designs (Adelekan et al., 2001; Lam et al., 2016), and the third was a pilot study (Veling et al., 2019).

In a study from Nigeria, Adelekan et al. (2001) assessed changes in TMHPs' mental healthcare knowledge, practice and attitudes after attending training sessions comprising modules on mental illness, treatment and aftercare. Two months after the training, TMHPs demonstrated significant improvements in the recognition of subtle, yet important symptoms, including undue sadness and

(a)

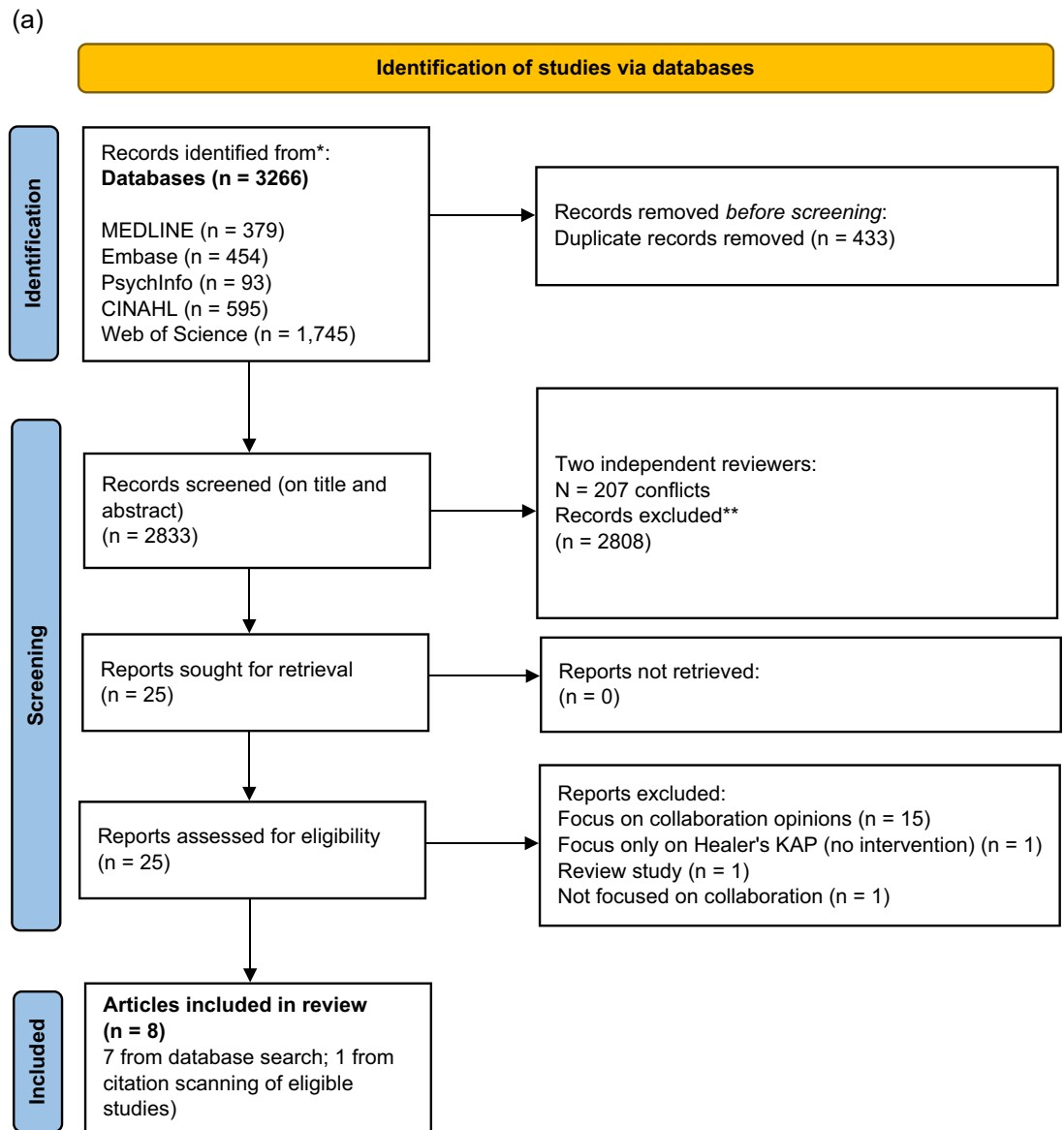

**Figure 1a.** PRISMA flowchart showing selection of studies from database inception to March 2023.

withdrawal. Further, they reported significant reductions in beating as a form of treatment and increases in the use of occupational therapy as an adjunct to treatment. The study suffered from considerable attrition with just 27/43 TMHPs completing the follow-up assessments.

In a second pre-post study from Hong Kong, Lam et al. (2016) delivered a 10-session Western mental health training course to TCM practitioners. Post training, confidence in recognising patients with psychological problems rose from 62.9% to 89.4%; diagnosing common mental health issues rose from 47.7% to 77.5%; and managing mental health problems rose from 31.2% to 64.3%. In qualitative responses, TCM practitioners observed how modern and traditional approaches might work in tandem and their role in this partnership: "Diagnosis of mental health problems and the side effects which occur after taking [Western medicine] pills. I realise we can give herbs or acupuncture to decrease [side effects] and make patients feel better" *(p.3)*.

Veling et al. (2019) conducted a pilot study to train 50 (out of a possible 200 in the area) TFHs to identify and refer recent onset

psychosis cases as part of a study on the incidence, course and treatment of psychotic disorders in a rural South African community. In addition to engaging with TFHs to develop a "mutual understanding" of traditional and biomedical concepts of psychosis, they developed a method for screening and referral for TFHs. Over a 6-month period, TFHs referred 149 clients with suspected recent-onset psychosis to the research team. The positive predictive value (PPV) of the TFHs' "disturbed" rating was 53.8% compared to a PPV of just 17.2% for those rated as "maybe disturbed." The authors concluded that TFHs can recognise recent-onset psychosis, though a full evaluation (including specificity and sensitivity of referrals) was not possible in this preliminary study.

More recently, Ben Zeev et al. (2024) used a mobile app to provide brief psychosocial interventions to healers, to encourage them to maintain human rights in their practice and to prompt them to monitor the status of their patients. The psychoeducation provided included guided relaxation techniques, rapport building, verbal de-escalation, challenging dysfunctional beliefs about psychiatric symptoms and protecting the human rights and dignity of

(b)

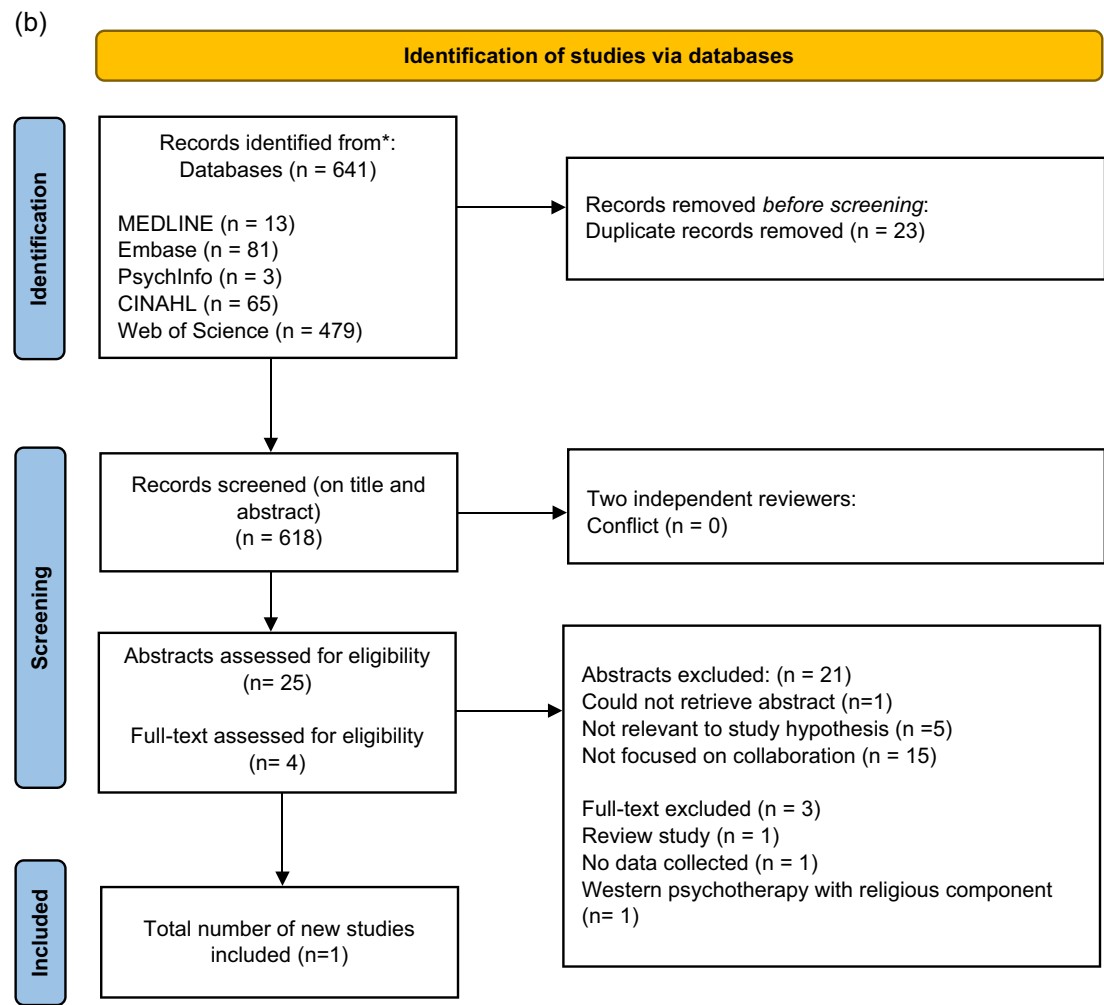

**Figure 1b.** PRISMA flowchart showing our updated search and selection of studies from March 2023 to December 2024.

patients. The intervention was delivered as brief digital animations or audio recordings with easy access to all psychoeducation contents. Overall, the authors reported a significant and clinically meaningful reduction in psychiatric symptom severity, psychological distress and shame at post treatment. Participants reported significantly reduced internalised stigma regarding their mental health conditions post treatment. Importantly, authors also reported a significant reduction in days chained at post treatment.

### (2) Shared collaborative models between TFHs and biomedical professionals

Three studies assessed shared collaborative models of care. In the first of two randomised controlled trials (COSIMPO study; (Gureje et al., 2020) tested the effectiveness of a manualised collaborative care model for patients with psychotic disorders in Ghana and Nigeria. The intervention involved TFHs and primary healthcare workers (PHCWs) working together to provide care for people admitted to the facilities of the TFHs. The PHCWs provided clinical support to respond to the medical (psychotic and physical) needs of the patients and to improve service through interactions with the TFH, patient and caregivers. The control condition comprised enhanced care as usual provided by the TFH (e.g., herbs, rituals, prayer). Due to ethical considerations, TFHs in both the intervention and control groups received training which included

information on the dangers of harmful practices and how to avoid them. Patients in the intervention group experienced significantly greater improvements in psychosis symptoms and evidenced significantly less disability compared to the control group. Both intervention and control groups experienced significant reductions in harmful practices.

The second RCT examined the efficacy of combining a psychotropic drug intervention with faith healing in a prayer camp in Ghana over a 6-week period (Ofori-Atta et al., 2018; Yaro et al., 2020). Mindful of ethical challenges, the researchers made efforts to reduce human rights abuses through education of staff and case-by-case reviews and comments to ensure that the study provided benefits to all residents in the sanatorium. At 6 weeks, patients in the experimental group (psychiatric care plus prayer camp treatment) reported significantly lower psychiatric symptoms compared to those receiving prayer camp treatment alone. However, there was no significant difference in the number of days in chains in either group (hours in chains were not measured). In a qualitative evaluation of this trial (Yaro et al., 2020), traditional healers reported enhanced knowledge about mental health and illness, human rights and increased collaboration between formal and informal healthcare providers: "The training was very helpful. It increased my knowledge about mental illness and the need to collaborate with hospital" (p. 4) (Yaro et al., 2020).

**Table 1.** Overview of study characteristics, intervention components, and main results

| Author/year | Country | Study design | Main assessments | Sample | Intervention description (educational or shared collaborative model) | Main results |
|---|---|---|---|---|---|---|
| Adelekan et al. (2001) | Nigeria | Pre-post intervention design | ■ Questionnaires on mental health knowledge, practice and attitudes administered before and after the training (following a 2-month free practice period) | 43 TMHPs: $M_{age}$ = 50.2 (SD = 15.5; male = 88%; female = 12%) 27 completed post intervention assessment: $M_{age}$ = 51 (SD = 14.4; male = 85%; female = 15%) | **Educational programme for TMHPs** Training sessions to improve mental health knowledge and practice including: ■ Concept of normality and abnormality ■ Types of mental illness ■ Treatment of mental illness ■ Aftercare/relapse prevention ■ Primary preventative measures ■ Introductory talks on sub-specialties | ■ TMHPs showed significant improvements ($p < .05$) in recognition of undue sadness, withdrawal and elation ■ TMHPs showed non-significant reductions in supernatural illness attributions (i.e., "curse") ■ Post-intervention, TMHPs reported that they no longer beat their patients as a form of treatment ■ Significant increase in the number of TMHPs using occupational therapy as an adjunct to treatment ($p < .05$) ■ Significant increase ($p < .05$) in number of TMHPs who claimed to practice regular follow-up |
| Gureje et al. (2020) | Nigeria Ghana | Cluster randomised controlled trial (RCT) | ■ PANNS, ISMI, WHO-DAS assessed at 3- and 6-month follow-up. ■ Harmful treatment practices assessed at 3 and 6 months. | Patients with active psychotic symptoms (PANNS ≥ 60) 166 intervention group: $M_{age}$ = 33.2 (SD = 12.1) 141 control (CAU by TFH) group: $M_{age}$ = 33.4 (SD = 10.2) | **Manualised collaborative shared care model for psychosis between TFHs and primary health care providers** In addition to training (for both groups) there were two main components of the intervention: ■ Clinical support to respond to the medical needs (e.g., psychotic or physical) of patients with psychosis. ■ Clinical support to improve service on a continuous basis through engagement with the TFH, patient and caregivers, focusing on reducing harmful treatment practices | ■ Patients in the intervention group had significantly greater improvements in positive, negative and general psychopathology ■ Patients in the intervention group had significantly greater improvements in course of illness and adjustment to work ■ Both groups experienced significant reductions in harmful practices (intervention: 57% to 9% vs control: 42% to 10%) ■ Intervention group had greater reductions in overall care costs (for total costs) |
| Lam et al. (2016) | Hong Kong | Pre-post intervention design | ■ Structured questionnaires designed for immediate pre-course and post-course | 151 TCM practitioners (age not reported; male = 42%; female = 58%) | **Educational programme for TCM practitioners** Comprising 10 interactive seminars within 3 months (2 hours per session) on topics relating to common psychological problems and psychotherapy including: ■ Overview and interview skills ■ Stress related disorders ■ Mood disorders including bipolar disorders ■ Somatoform disorders, panic and phobic disorders, obsessive-compulsive, and related disorders ■ Psychotherapy ■ Substance abuse including alcoholism ■ Psychotic disorders ■ Sleep disorders | After intervention, there was a significant increase in the proportion of TCM practitioners confident in: ■ Recognising patients with psychological problems (62.9% vs 89.4%; $p < .001$) ■ Diagnosing common mental health problems (47.7% vs 77.5%; $p < .001$) ■ Managing patients with common mental health problems (31.2% vs 64.3%; $p < .001$) ■ 66.9% perceived that their mental health care management had improved after attending the 10-session training course |

**Table 1.** (*Continued*)

| Author/year | Country | Study design | Main assessments | Sample | Intervention description (educational or shared collaborative model) | Main results |
|---|---|---|---|---|---|---|
| Ofori-Atta et al. (2018) | Ghana | Randomised controlled trial (RCT) | ▪ BPRS, GAF, BSI, PHQ | Patients with schizophrenia, bipolar disorder, and major depressive disorder 71 Intervention groups (psychiatric care and prayer camp treatment) 61 control group (usual prayer camp treatment) | **Collaborative model joining psychiatric care with faith healing in a prayer camp** Comprised: ▪ Prescription of clinically indicated medications by psychiatrist/senior medical officers ▪ Prayer camp treatment including a combination of prayer and Bible study and fasting for 3–21 days before participation in the study | ▪ Psychotic symptoms were significantly lower ($p$ = .003) in the experimental group ▪ Significantly higher functioning scores in the intervention group ($p < .0001$) ▪ There were no significant reductions in days in chains in intervention or control groups |
| Saha et al. (2021)/ Shields et al. (2016) | India | Mixed methods study (including qualitative interviews and secondary analysis) | Analysis of referral cases Interview guide developed by authors | Saha: 26 patients (9 Dava patients 8 Dua patients 9 Dava-Dua patients) 6 mental health service providers Shields: 3 AMHPs 3 FBHs 3 patients 7 carers | **Collaborative model combining psychiatric medicine ("Dava") and FBH ("Dua")** Comprised: ▪ AMHPs delivered an outpatient clinic within the dargah (shrine) including medication and basic counselling ▪ Mujavars delivered rituals and referred to AMHPs if they detected a mental health problem ▪ Programme included training for FBHs on mental illness, referral strategies, and the referral process | ▪ 7149 patients visited Dava-Dua between 2008 and 2018 ▪ Over a 5-year period (2008–2013) FBHs referred 57.9% of clients receiving care, though referrals from FBHs have declined over time (to 37%) ▪ Clients visiting the Dava Dua attributed their improvement to a combination of the rituals they completed with the FBHs and the medication and basic counselling they received from the AMHPs |
| Veling et al. (2019) | Rural South Africa | Pilot study | CAPE SCAN | 50 THPs 149 help-seeking clients referred by THPs | **Collaborative model to enhance THPs' screening and referral of individuals with recent onset psychosis** Programme comprised: ▪ Engagement with community leadership ▪ Establishment of a Community Research Advisory Board (CRAB) ▪ Engagement with THPs to develop mutual understanding of traditional and biomedical concepts of psychosis ▪ Development of a method for screening and referral by THPs | ▪ The positive predictive value of the THP "disturbed" rating was 53.8% |
| Yaro et al. (2020)* | Ghana | Qualitative evaluation (of an RCT) | In-depth interviews | 11 spiritual healers; 21 traditional medical practitioners; 13 patients and their carers and 9 CPNs | See description above (Ofori-Atta et al., 2018) | ▪ Training increased THPs' level of knowledge and understanding about mental conditions ▪ Participants reported increased collaboration between biomedical and traditional healthcare providers |
| Ben Zeev et al. (2024) | Ghana | Pre-post intervention design | BPRS, BSI, TBDI, OAS, Brief ISMI, PHQ, LQOLI, BMQ-General, WAI Days chained, days of forced fasting | 4 TFHs 17 patients $M_{age}$ = 44.3 (SD = 13.9) | **Collaboration model of Digital Educational programme for TFHs and pharmacotherapy from visiting nurse** | ▪ Significant and clinically meaningful reduction in psychiatric symptom severity, psychological |

**Table 1.** (*Continued*)

| Author/year | Country | Study design | Main assessments | Sample | Intervention description (educational or shared collaborative model) | Main results |
|---|---|---|---|---|---|---|
| | | | | Male = 59% Female = 41% | Digital Mobile Application for TFHs <br>■ Psychoeducation included: guided relaxation techniques, rapport building, verbal de-escalation, challenging dysfunctional beliefs about psychiatric symptoms, and preservation of human rights and dignity in practice <br>■ App allows healer to track and monitor progress of individual patients in the camp. The App prompts healers every day to check in with each patient and provide a rating <br>■ Daily psychosocial digital animation training videos for healers <br>■ Visiting community nurse provided pharmacological care directly to patients at the prayer camp | distress, shame and stigma <br>■ Authors reported a significant reduction in days chained and promising trend for reduction in days of forced fasting <br>■ The intervention seems to be feasible, acceptable, safe, and clinically promising. Preliminary findings suggest that the digital intervention may have shifted healers' behaviours at the prayer camp and committed fewer human rights abuses |

*Notes:* TMHP: traditional mental health practitioner; TFH: traditional faith healer; TCM: traditional Chinese medicine; PHCW: primary healthcare worker; AMHP: adult mental health practitioner; PANNS: positive and negative syndrome scale; ISMI: internalised stigma of mental disorder; WHO-DAS: WHO disability assessment schedule; BPRS: brief psychotic rating scale; BSI: brief symptom inventory; PHQ: patient health questionnaire; CAPE: community assessment of psychic experience; SCAN: schedules for clinical assessment in neuropsychiatry; TBDI: Talbieh brief distress inventory; OAS: other as Shamer scale; Brief ISMI: internalized stigma of mental illness; LQOLI: Lehman quality of life inventory; BMQ-General: beliefs about medications questionnaire – general harm subscale; WAI: working alliance inventory.
*Qualitative evaluation of Ofori-Atta study.

**Table 2.** An outline of key outcomes in studies, and their potential mechanisms and contextual moderators

| Study | Key outcomes | Proposed mechanisms underpinning outcomes | Contextual barriers | Contextual facilitators |
|---|---|---|---|---|
| Adelekan et al. (2001) | ■ Widening recognition of mental health symptoms <br>■ Reduction in the habit of beating patients <br>■ Greater adoption of standard practices | ■ Increased knowledge/awareness <br>■ Change in attitudes and beliefs | ■ Suspicion from some TFHs <br>■ Limited funds available for research | ■ High level of co-operation from TFHs <br>■ Mutual understanding of modern and traditional practices |
| Gureje et al. (2020) | ■ Reductions in psychotic symptoms <br>■ Reductions in harmful treatment practices, for example, shackling | | | ■ Incentives for providers <br>■ Free medications for the trial |
| Lam et al. (2016) | ■ Increased confidence in recognising patients with psychological problems <br>■ Increased intention to refer (but not supported by referral rates) | ■ Increased awareness/better understanding of mental disorders and management <br>■ Increased confidence <br>■ Collaborative learning approach, for example, case sharing | ■ Difficulties in understanding medical terms <br>■ Consultation time constraints <br>■ Lack of formal referral systems <br>■ Patients' negative attitudes | ■ Open minded attitudes of teachers <br>■ Involving TCM practitioners as tutors (suggested facilitator) |
| Ofori-Atta et al. (2018)/Yaro et al. (2020) | ■ Reduction in psychiatric symptoms <br>■ Reduction in harmful practices, for example, days in chains | ■ Enhanced knowledge about mental health and illness and human rights <br>■ Increased collaboration between orthodox medical practitioners and traditional/spiritual healers | ■ Potential incomplete integration of medical team into decision making by prayer camp staff – co-location rather than full integration | ■ Provision of incentives, for example, health insurance available to all sanatorium residents, making the camp a recognised model |

(*Continued*)

**Table 2.** (*Continued*)

| Study | Key outcomes | Proposed mechanisms underpinning outcomes | Contextual barriers | Contextual facilitators |
|---|---|---|---|---|
| | ■ Increased belief in bio-medical approaches by TFHs | | ■ Belief in religious not bio-medical model of mental illness<br>■ Shortage of psychiatric medications | ■ Creating an atmosphere of mutual understanding through respectful exchange of ideas |
| Saha et al. (2021)/Shields et al. (2016) | ■ Improvements in patients' awareness of mental illness and belief in benefits of psychiatric medication<br>■ Improvement in mental health literacy for FBHs including reconceptualisation of clients' problems | ■ Building rapport and trust (e.g., continuous and open dialogue to promote mutual understanding, develop unified goals based on common values, supporting rather than condemning FBHs)<br>■ Empowerment of FBHs through training and sensitisation activities<br>■ Highlighting complementary aspects of both systems<br>■ Mutual referral<br>■ Redefining the roles of AMHPs and FBHs | ■ Apprehension of professionals in both systems<br>■ Perceived differences in professional and societal status between biomedical and FBHs<br>■ Reluctance of AMHPs to equalise their status with FBHs | ■ Free cost of treatment to alleviate financial burden<br>■ Cross-referrals enabling FBHs to maintain their income (suggested reason that FBHs overcame their initial resistance to collaborating with AMHPs) |
| Veiling (2019) | ■ Referral of recent onset psychosis cases by THPs | ■ Trust building (through long term engagement and mutual respect)<br>■ Common understanding of psychiatric concepts | | ■ Recognising and acknowledging local authorities<br>■ Taking time to develop relationships<br>■ Adaptation of procedures to socio-cultural norms |
| Ben Zeev et al. (2024) | ■ Reduction in psychiatric symptom severity, psychological distress and shame<br>■ Reduced internalized stigma regarding mental health conditions<br>■ Significant reduction in days chained | ■ Increased knowledge about mental health (i.e., challenging beliefs about psychiatric symptoms, human rights, and psychosocial interventions (i.e., rapport building)<br>■ Increased collaboration between healers and medical practitioners | | ■ Mobile app providing easily accessible psychoeducation in the form of brief digital animations or audio recordings<br>■ Mobile app prompting healers to interact with psychoeducation materials<br>■ Mobile apps allowed healers to create a list of active patients to support basic tracking and monitoring of individual patient progress<br>■ Both healer and patient participants were compensated for their involvement in the study |

In a multi-method study (i.e., secondary analysis of case records; qualitative interviews) from Gujarat, India, researchers developed a collaborative model of mental healthcare comprising modern medicine ("Dava") and traditional faith healing (prayer: "Dua") (Shields et al., 2016; Saha et al., 2021). FBHs from the Mira Datar *dargah* (shrine) and allopathic mental health practitioners (AMHPs) worked together in partnership to deliver mental healthcare to the rural community. AMHPs started a psychiatric outpatient clinic in the *dargah* where FBHs treated patients with rituals. FBHs referred patients they suspected to have mental health problems to the psychiatric clinic for diagnosis, treatment and counselling. Equally, AMHPS could refer patients back to the FBHs if they felt problems could be addressed through spiritual rituals.

Clients with more severe mental health problems were referred to the government run psychiatric hospital in the city. A total of 7149 patients visited the Dava-Dua centre between July 2008 and March 2018. Over a 5-year period (2008–2013), FBHs referred 57.9% of clients receiving care; however, referrals from FBHs have declined over time to 37%, while referrals from friends and relatives have increased (Saha et al., 2021). Qualitative interviews indicated an appreciation for a holistic approach within the Dava-Dua: "I had a perception that … people get cured only by getting medicines. But

once I started working here, I realized that it was not only the medicines working, but it is the faith and support of others which is making it work" (AMHP, p.382).

Ben Zeev et al. (2024) included a mobile nurse alongside their mobile app intervention to provide pharmacotherapy to monitor patients at a prayer camp in Ghana. At the initial visit to the prayer camp, 15 participants consented to receiving pharmacotherapy. The nurse was able to assess, provide pharmacotherapy to and monitor patients weekly. Overall, 110 medication follow-up visits were conducted by the mobile nurse. The intervention proved to be safe and helped to promote better care in the prayer camp (i.e., some participants were referred to the district hospital as they were identified as requiring immediate medical attention).

### Potential mechanisms and contextual moderators of intervention outcomes

Table 2 outlines proposed mechanisms and contextual barriers and facilitators underpinning intervention outcomes.

The proposed mechanisms underpinning successful collaboration included building trust, respect and rapport (Shields et al., 2016; Veling et al., 2019); empowering TFHs (Lam et al., 2016;

Shields et al., 2016) by increasing awareness and knowledge of mental health problems and human rights (Adelekan et al., 2001; Lam et al., 2016; Ofori-Atta et al., 2018; Ben Zeev et al., 2024); highlighting the complementary aspects of both modern and traditional systems (Shields et al., 2016) and cultivating mutual understanding and unified goals through a collaborative approach (Lam et al., 2016; Shields et al., 2016). For instance, Adelekan supported a group of healers through a comprehensive training program which increased their knowledge, attitudes and practice.

Moderators (barriers and facilitators) of outcomes were observed at the individual, relationship and service levels (Yaro et al., 2020). For instance, reductions in psychiatric symptoms and harmful practices were potentially driven by knowledge about mental health and illness and human rights and through an increased collaboration between medical practitioners and healers by creating an atmosphere of mutual understanding through respectful exchange of ideas.

With regard to barriers, TFHs were suspicious of biomedical practitioners and felt that they posed a threat to their livelihood (Adelekan et al., 2001; Saha et al., 2021). Biomedical practitioners, in turn, were apprehensive of working with TFHs due to differences in perceived status, and some were reluctant to equalise their status with TFHs (Shields et al., 2016). Studies noted a gap in the understanding of mental illness and associated terms (i.e., religious versus biomedical understanding) (Lam et al., 2016; Ofori-Atta et al., 2018). In terms of service level barriers, participants highlighted a lack of formal referral systems for TFHs to refer to biomedical practitioners (Lam et al., 2016), limited time available to understand the patient's background (Lam et al., 2016), incomplete integration of medical teams within the traditional setting (Ofori-Atta et al., 2018) and a shortage of psychiatric medications (Yaro et al., 2020).

Several contextual facilitators were identified throughout the studies including (1) the provision of incentives (e.g., health insurance available to all sanatorium residents) (Shields et al., 2016; Yaro et al., 2020; Ben Zeev et al., 2024). In studies where incentives were provided, there also appeared to be improvements in patient mental health symptoms; (2) creating an atmosphere of openness (e.g., not a Western dominant attitude) (Lam et al. 2016; Yaro et al., 2020) led to increased confidence in healers ability to recognise mental health conditions in patients and increased belief in biomedical services; (3) the adaptation of procedures to socio-cultural norms (e.g., pathways to care, treatment options, explanatory models and idioms of distress) (Veling et al., 2019) leading to increased referral to biomedical services by healers.

Moreover, deploying modern mobile methods for collaboration highlights the importance of using technology for prompting (i.e., reminding healers to check the mental health status of their patients), providing daily training/information via an app ensuring that the method of learning is appropriate to modern day life of healers and allows them to access the information where and when they want. As Ben Zeev et al. (2024) showed, their dual pronged intervention of providing psychoeducation through an app and supporting pharmacological treatment with a mobile nurse at a prayer camp in Ghana, significantly reduced the severity of psychiatric symptoms, psychological distress, feelings of shame and stigma, alongside a reduction in harmful practices (i.e., chaining and forced fasting).

## Discussion

We completed a systematic scoping review of studies investigating the outcomes of collaborative models between biomedical and traditional practitioners. We identified very few studies, and there were only two using randomised controlled methods. Broadly speaking, we identified two main approaches: those comprising training or educational programmes for traditional healers and those combining biomedical and traditional approaches in a collaborative care model.

Reflective of methodological approach, we found that shared collaborative models demonstrated the strongest evidence. Two RCT studies reported significantly greater improvements in psychiatric symptoms for patients receiving the intervention (biomedical plus traditional approaches) compared to those receiving the control treatment (traditional approaches alone) (Ofori-Atta et al., 2018; Gureje et al., 2020). What we cannot ascertain from these two studies is how the intervention would have compared to biomedical treatment alone. A non-experimental mixed methods study also indicated that collaboration between modern medicine and faith-based treatment can benefit patients, especially those with limited access to mental healthcare (Saha et al., 2021). Collaborative models shared commonalities including the administration of psychiatric medication and counselling (Shields et al., 2016) within a traditional setting (i.e., TFH facilities, prayer camp, shrine) and the provision of training, education and/or supervision for TFHs. As TFHs are often the first point of contact for people in LMICs (especially rural areas) (Singh et al., 2023), locating collaborative models within traditional settings appears key to the success of these programmes.

There were no significant differences in reductions in harmful practices between intervention and control groups in both RCTs. In one of the RCTs, harmful practices were significantly reduced in *both* intervention and control groups (Gureje et al., 2020). Due to ethical concerns, TFHs were trained and closely monitored in both control and intervention groups in this study. This indicates that healers can be trained (and supervised) to reduce the use of harmful practices. This observation is bolstered by the findings of Adelekan et al. (2001) in which an educational program on mental health knowledge and practice for faith healers led to a significant reduction in (self-reported) beating as a form of treatment. It is curious that there was no reduction in chaining in the second RCT following the intervention (Ofori-Atta et al., 2018). Potential reasons included incomplete integration of the medical team into decision-making, the need for more intensive training on human rights and the dangers of harmful practice and lack of sensitivity of outcome measures (i.e., days in chains rather than hours in chains measured) (Ofori-Atta et al., 2018). As harmful practices within faith-based approaches are viewed as one of the main barriers to collaboration between biomedical and faith-based services (Shields et al., 2016), more work is needed to understand what might enhance or impede a rights-based approach by TFHs, including collaboration rather than condemnation of FBHs (Shields et al., 2016).

Our exploration of the potential mechanisms and moderators (i.e., facilitators and barriers) of intervention effects highlighted several important considerations for researchers when developing collaborative approaches. First, when developing collaborative models or educational programmes, it is crucial to invest time in building rapport and establishing trust with communities and their leaders (Veling et al., 2019; Saha et al., 2021). Across studies, it was evident that building trust is a pre-requisite for collaboration, highlighting the importance of cultural sensitivity and mutual respect (Van der Watt et al., 2018) to facilitate the integration of modern and traditional approaches rather than a "co-location" of approaches (Ofori-Atta et al., 2018). Previous qualitative work has indicated that traditional healers feel demeaned by clinicians who

disregard their mode of treatment and stereotype them as "dirty" (Musyimi et al., 2016). This highlights the importance of dialogue formation between biomedical and traditional practitioners with a consideration of facilitators (e.g., protection of traditional medicine, compensation of healers, education of both groups, adequate community involvement) to enhance sustainability (Musyimi et al., 2016). Other key barriers included service infrastructure (including lack of formal referral systems) and limited resources (e.g., lack of psychiatric drugs on national health insurance), which impeded joint work within the community (Yaro et al., 2020). Recently, the WHO renewed their commitment to incorporate traditional healers in the provision of healthcare (Eurocam 2024), providing impetus for increasing funding for collaborative models in mental healthcare in places such as Africa (Yaro et al., 2020). We highlight the recent work by Ben Zeev et al. (2024) who navigated these challenges by developing a mobile app which provided psychoeducation (i.e., rapport building, guided relaxation techniques) to healers at a prayer camp in Ghana, in an accessible and contextually appropriate manner. Furthermore, the app allowed healers to monitor and follow up their patients, demonstrating an innovative way of supporting healers in their practice and improving patient well-being.

### Limitations

Our review has limitations. First, we were only able to identify a very small number of studies, and these were mostly located in Africa. Thus, the generalisability of our results to other countries is unclear. There were some indications that educational models could also be effective (in increasing knowledge and improving practice) in China and India; however, more work is needed in these (and other low- and middle-income) countries, especially in view of the importance of cultural adaptations in low resource settings (Veling et al., 2019). Second, there were only two RCT studies, which means results should be viewed as preliminary. Further, these studies compared interventions to traditional approaches only (rather than also comparing to modern approaches only). Other study designs included a pilot study, pre-post studies and qualitative/descriptive work. These studies had a number of methodological limitations, including small sample size (Veling et al., 2019), exacerbated by attrition (Adelekan et al., 2001), potential self-report bias (Adelekan et al., 2001; Lam et al., 2016; Gureje et al., 2020), relatively low percentage recruitment of faith healers (Veling et al., 2019), incomplete evaluation of the programme (Veling et al., 2019) and lack of clarity/systematic evaluation of treatment approach of healers (e.g., lack of clarity on how many patients were seen by healers before referring (Veling et al., 2019) and limited details on treatment modalities of healers (Gureje et al., 2020). This restricts our ability to fully establish effectiveness, potential mediators and contextual moderators of intervention effects (Breitborde et al., 2010). Third, we only included English language studies in our review, which could have excluded some programmes. Furthermore, future studies should examine the impact of different healthcare systems on typologies of collaboration. In addition, prospective studies should endeavour to provide an in-depth cost analysis to fully understand the feasibility and sustainability of collaboration efforts.

### Conclusions

Combining modern and traditional approaches to mental healthcare appears to be a promising approach to help reduce the mental health gap by providing more accessible care to people in low-

resource settings (Singh et al., 2022; Bhogal et al., 2024). These approaches reflect a patient-centred orientation, offering a more personalised and holistic spectrum of care blending both traditional and biomedical practices (Shields et al., 2016). Moving forward, research programmes should consider including active participation from stakeholders (e.g., patients and their caregivers, healers, community health workers) to explore community understandings of serious mental disorders (SMDs) and help-seeking and perspectives on faith-based healing for SMDs (Singh et al., 2022). Other work should consider how we can enhance the adoption and sustainability of collaborative models at scale (Gureje et al., 2020). For instance, the promising findings from Ben Zeev et al. (2024) illustrate a creative way of engaging healers by using easily accessible digital tools that have the potential to enhance the adoption of collaborative models and achieve scalability and sustainability. Due to the scarcity of mental health professionals in LMICs, government investment in such technologies can address the significant shortages in LMICs while simultaneously significantly improving patient outcomes on a large scale. Aside from increasing government funding, additional work with policymakers should include increasing formal recognition and regulation of faith healers, developing strategies to reduce deep distrust and feeling of superiority between paradigms (e.g., co-design of collaborative models; Singh et al., 2022) and increasing mutual understanding and shared responsibility for patient well-being (Van der Watt et al., 2018).

**Open peer review.** To view the open peer review materials for this article, please visit http://doi.org/10.1017/gmh.2025.20.

**Supplementary material.** The supplementary material for this article can be found at http://doi.org/10.1017/gmh.2025.20.

**Data availability statement.** Data are available upon request to the corresponding author.

**Acknowledgements.** Members of the TRANSFORM consortium include the Asido Foundation (https://asidofoundation.com/, Ibadan, Nigeria), Bangladesh Center for Communication Programs (BCCP, https://www.bangladesh-ccp.org/, Dhaka, Bangladesh) and the External Scientific Advisory Board (ESAB): Prof. Nahid Mahjabin Morshed (Department of Psychiatry, Bangabandhu Sheikh Mujib Medical University), Prof. Vincent Agyapong (Department of Psychiatry, Dalhousie University), Prof. Jude Uzoma Ohaeri (Faculty of Medical Science, University of Nigeria), Dr Helen Liebling (Coventry University) and Dr Vandana Gopikumar (The Banyan and The Banyan Academy in Leadership for Mental health, Chennai, India). The site PIs are Prof Olayinka Omigbodun (Ibadan) and Dr Tanjir Soron (Dhaka). We would like to acknowledge the wider TRANSFORM team who will be working on this study. They include Dr Henry Nwankwo, Simon Smith, Dr Kafayat Aminu, Ms. Oluwatosin Obuene, Dr Tolulope Bella-Awusah, Ms. Adeola Oduguwa (Afolayan), Ms. Olubukola Falola, Ms. Bukola Arowojolu, Mr. Tomisin Ogunmola, Ms Nadia Binte Alam, Dr Azmery Kazi Shammin, Dr Maksudar Rony, Dr Rana Zaman and Dr Farzana Ahmed.

**Author contribution.** SPS is the PI of the TRANSFORM study. SJ and SAJ designed the search strategy and ran the searches. SJ, DF, OI and CW reviewed the articles against the inclusion and exclusion criteria and extracted data. SJ and CW wrote the manuscript. SS, DM, RL and SPS supported writing of the manuscript.

**Financial support.** This study/project is funded by the UK's National Institute for Health and Care Research (NIHR) (Award number: NIHR200846). SPS is supported by the NIHR Applied Research Collaboration (ARC) West Midlands. The views expressed are those of the author(s) and not necessarily those of the NIHR or the Department of Health and Social Care.

**Competing interest.** The authors have no conflict of interest.

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
