## [Reviewer Report]

REVIEW : Cambridge Prism

A Scoping Review to Evaluate the Efficacy of Combining Traditional Healing and Modern Psychiatry in Global Mental Health Care

Overview

This is an important area of study given that the majority of patients in LMIC consult traditional herbalists, divinationsits and faith healers for assessment, diagnosis and treatment of not only mental disorders, but also physical conditions. Often this is the most accessible, reachable, and affordable treatments. Apart from this, they are more acceptable as they offer answers as to why these conditions are affecting the particular individual. The why questions are important in some of the cultures and yet are not satisfactorily answered in the biomedical approaches. Research has thus been focused on integration of these practices in the current biomedical approaches.

Abstract

The abstract is written well.

INTRODUCTION

The rationale for the study is clear and is well presented. However, it is important to separate the diviners, herbalists and faith healers. This is especially so as the herbalists use treatments that are potentially psychoactive and whose pharmacological actions have not been well elucidated.

METHODS

Eligibility criteria

This section needs to be revised. The authors use the future tense, when it should be in the past tense.

RESULTS

The results section is written well and represents the findings. Line 235-236 does not read well, please revise.

DISCUSSION

The authors were limited by the paucity of studies in the field and the conclusions drawn are thus very inconclusive. The studies reviewed are also affected by the fact that the traditional, faith and herbalists are lumped together, with no due recognition that the herbalists in particular may have formularies that maybe pharmacologically active and may produce their effects from that.

In sentence 377-379, Africa is referred to as a country, it is a continent and the authors need to correct that.

---

## [Reviewer Report]

Jilka et al provide a well written description of a scoping review of the global literature on collaborative mental health care models including traditional and faith healers, and approach to delivering mental health care that is understudied despite having potential to substantially contribute to closing the mental health treatment gap. I have two major concerns. For one, it was unclear to me whether care models for people living with severe mental illness (SMI) were the focus of this review or not. The title suggests this review is about care models for all mental illness and the methods do not seem to indicate restriction related to type of mental illness, but the framing of the paper seems to indicate the interest is on treatment of SMI by healers (e.g., the significance in the introduction, focus on data related to healers using harmful practices associated with SMI). Second, the search goes up through March 2023, over one year ago. I suggest the authors re-run the search to ensure no additional papers have been published between their initial search and this manuscript submission. Below, I detail additional, more minor comments:

Methods:

“We will exclude studies focusing on people with lived experience and/or caregivers, and other

stakeholders.” It is not clear what this means. One would assume that participants in studies evaluating an intervention include patients, i.e., people with lived experience. I believe this exclusion criteria is in relation to the “provider”, but I believe the authors should clarify. Same for the following sentence, “We will exclude the viewpoint or outcomes from the perspective of the persons with lived experience and their caregivers.” As change in psychiatric symptoms is evaluated in the present study, and presumably these symptoms were from the perspective of a patient, this sentence is confusing.

“However, studies will be excluded if traditional healers provided any oral or topical or nasal or inhaling herbal/ chemical/ substances for the management of common mental illness.” Why was this an exclusion criterion?

Line 150: I believe the authors mean to say “duplicate datA” instead of “duplicate datE”.

I think the inclusion of potential mechanisms and moderators is a very interesting component of this paper. However, I think the methodology for extracting moderator data and it’s presentation could use more clarification. How did the authors determine what were barriers and facilitators? Determinants stated by the authors? Determinants derived from qualitative and or quantitative data? With the potential mechanisms data, Table 1 and Table 2 can be crossed checked to determine what methods led to the identification of a mechanism in each paper. This is not clear for barriers and facilitators. The source of moderator data may not be able to be clarified in table format, but could potentially be detailed more in the results.

Limitations: LAMI is used without a definition. I would suggest just writing out low and middle income and not using an acronym.

---

## [Reviewer Report]

Key Review Questions:

1. Global Content Coverage:

- The review includes studies from multiple areas, but most are from Africa (4 out of 6 independent studies). The discussion acknowledges this geographical limitation

- 6 studies for a scoping review is very limited. It could be helpful to include qualitative/descriptive studies and grey literature given these limitations.

2. Global Context Integration:

- The authors effectively situate findings within global mental health challenges

- Good discussion of implications for LMICs broadly

- Clear connection to WHO policies and global mental health priorities

- Strong consideration of cultural adaptation needs across different settings

Strengths:

1. Methodology:

- Clear systematic scoping review methodology

- Well-defined inclusion/exclusion criteria

- Transparent search and screening process

- Good data extraction and synthesis approach

2. Analysis:

- Comprehensive analysis of intervention types

- Clear presentation of outcomes

- Thoughtful analysis of mechanisms and moderators

- Good balance of quantitative and qualitative evidence

3. Presentation:

- Well-structured manuscript

- Clear tables summarizing findings

- Good flow from background to conclusions

- Appropriate use of examples and quotes

Areas for Improvement:

1. Geographic Coverage:

- Could discuss strategies for expanding evidence base beyond Africa

- More analysis needed of why certain regions are underrepresented

- Consider implications of geographical skew for generalizability

2. Methodological:

- Could provide more detail on quality assessment of included studies

- Risk of bias assessment could be more explicit

- More discussion of limitations of included studies needed

3. Analysis:

- Could strengthen analysis of contextual factors across settings

- More detailed comparison across different healthcare systems

- Deeper analysis of cost implications and sustainability

4. Implementation:

- Could provide more concrete recommendations for practice

- More discussion of scaling up successful interventions

- Additional analysis of policy implications

Overall Assessment: This is a well-conducted scoping review that makes an important contribution to understanding collaborative models between traditional and modern mental healthcare in LMICs. However, the number of studies reviewed is a significant limitation and there is a lack of quality assessment of the sources.